# Angiotensin-(1-7)—A Potential Remedy for AKI: Insights Derived from the COVID-19 Pandemic

**DOI:** 10.3390/jcm10061200

**Published:** 2021-03-13

**Authors:** Samuel N. Heyman, Thomas Walther, Zaid Abassi

**Affiliations:** 1Department of Medicine, Hadassah Hebrew University Hospital, Mt. Scopus, Jerusalem 91240, Israel; 2Department of Pharmacology and Therapeutics, School of Medicine and School of Pharmacy, University College Cork, T12 YN60 Cork, Ireland; t.walther@ucc.ie; 3Institute of Medical Biochemistry and Molecular Biology, University Medicine Greifswald, 17489 Greifswald, Germany; 4Department of Physiology and Biophysics, Rappaport Faculty of Medicine, Technion-Israel Institute of Technology, Haifa 3200003, Israel; abassi@technion.ac.il; 5Department of Laboratory Medicine, Rambam Health Campus, Haifa 3109601, Israel

**Keywords:** COVID-19, acute kidney injury, angiotensin 1-7, Mas receptor, ACE2, RAAS

## Abstract

Membrane-bound angiotensin converting enzyme (ACE) 2 serves as a receptor for the Sars-CoV-2 spike protein, permitting viral attachment to target host cells. The COVID-19 pandemic brought into light ACE2, its principal product angiotensin (Ang) 1-7, and the G protein-coupled receptor for the heptapeptide (MasR), which together form a still under-recognized arm of the renin–angiotensin system (RAS). This axis counteracts vasoconstriction, inflammation and fibrosis, generated by the more familiar deleterious arm of RAS, including ACE, Ang II and the ang II type 1 receptor (AT1R). The COVID-19 disease is characterized by the depletion of ACE2 and Ang-(1-7), conceivably playing a central role in the devastating cytokine storm that characterizes this disorder. ACE2 repletion and the administration of Ang-(1-7) constitute the therapeutic options currently tested in the management of severe COVID-19 disease cases. Based on their beneficial effects, both ACE2 and Ang-(1-7) have also been suggested to slow the progression of experimental diabetic and hypertensive chronic kidney disease (CKD). Herein, we report a further step undertaken recently, utilizing this type of intervention in the management of evolving acute kidney injury (AKI), with the expectation of renal vasodilation and the attenuation of oxidative stress, inflammation, renal parenchymal damage and subsequent fibrosis. Most outcomes indicate that triggering the ACE2/Ang-(1-7)/MasR axis may be renoprotective in the setup of AKI. Yet, there is contradicting evidence that under certain conditions it may accelerate renal damage in CKD and AKI. The nature of these conflicting outcomes requires further elucidation.

## 1. Introduction

Acute kidney injury (AKI) remains a principal cause of morbidity and mortality among hospitalized patients, and plays an important role in the initiation and progression of chronic kidney disease (CKD). We now better understand AKI as a heterogeneous disease entity with diverse etiologies and pathophysiologies, including tubular injury generated by hypoxia, oxidative stress, direct cytotoxicity and inflammation [1]. However, most interventions aimed at the prevention or amelioration of evolving AKI so far have failed, perhaps, in part, because nearly all strategies and innovations were tested using flawed experimental animal models that do not replicate the relevant clinical scenario [1,2].

Angiotensin (Ang) 1-7, mainly generated by the angiotensin converting enzyme (ACE)2, until recently a disregarded axis of the renin–angiotensin system (RAS) [3], exerts renal vasodilation and attenuates inflammation, oxidative stress, apoptosis and fibrosis. Although known about for over 20 years, the interest in Ang-(1-7) and its downstream signaling through the G protein-coupled receptor (MasR) has further surged during the Sars-CoV-2 pandemic, since many features of the cytokine storm and coagulopathy characterizing the COVID-19 disease may be linked to the depletion of the ACE2 and Ang-(1-7)-mediated downstream signal in vital organs, along the unleashed Ang II/AT1R axis [4,5]. This review outlines the generation and role of the ACE2/Ang-(1-7)/MasR axis in balancing the devastating actions of the more familiar Ang II-triggered pressor arm of RAS, and delineates the disrupted Ang II/Ang-(1-7) balance that is generated in the COVID-19 disease. We shall discuss the alterations and the well-established renoprotective impact of the ACE2/Ang-(1-7)/MasR axis, attenuating the progression of CKD, and with the same rationale describe preliminary animal studies indicating that the stimulation of the ACE2/Ang-(1-7)/MasR axis may serve as a novel therapeutic option in COVID-19 patients, with special focus on the management of evolving AKI.

## 2. The RAS: Counteracting Harmful and Protective Pathways

The RAS has been gradually revealed over the last 120 years [3,6]. The general initial concept, consolidated by the end of the previous century, was that of a paracrine/endocrine system, responsible for salt retention and vasoconstriction, with consequent hypertension and the induction of additional injurious pathways, including the promotion of inflammation, oxidative stress, cell proliferation, apoptosis, coagulation and fibrosis [3,7]. Most adverse outcomes were attributed to angiotensin II (Ang II), generated from angiotensin I (Ang I) by the angiotensin-converting enzyme (ACE), and activating the Ang II type 1 receptor (AT1R) (Figure 1). Indeed, the introduction of ACE inhibitors (ACEi), and later of AT1R blockers (ARBs), was found to attenuate the progression of cardiovascular morbidities [8,9,10,11] and of declining kidney function in patients with CKD [12,13,14,15], with an evident reduction in the development of cardiac and renal fibrosis [16,17,18]. Aldosterone, a second and terminal messenger of Ang II, was also found to independently hasten organ fibrosis, a process amended by aldosterone antagonists [19].

Within the last 30 years, we have learned that the RAS system is much more complex, and that, in addition to the deleterious ACE/Ang II/AT1R axis (the “pressor arm”), there are inherent counterbalancing and protective mechanisms that attenuate vasoconstriction and the adverse effects of the already familiar ACE/Ang II/AT1R axis [20]. Step-by-step, new protective components of the RAS axis (the “depressor” or protective arm) have been uncovered; first the Ang II type 2 receptor (AT2R), activated by Ang II, and later the ACE2/Ang(1-7)/MasR axis, revealed principally by Santos et al. [21]. ACE2 is a peptidase located on the cell membranes in various tissues, including the kidneys. As illustrated in Figure 1, it promotes the proteolytic cleavage of the octapeptide Ang II with the formation of Ang-(1-7). Alternatively, Ang 1-7 can be generated by a preliminary cleavage of the decapeptide Ang I with the formation of Ang 1-9, followed by the removal of two additional amino acids by ACE. Ang 1-7 may also be formed directly from Ang I, following proteolytic cleavage by neprilysin, and other endopeptidases [3,22,23].

As opposed to the hazardous consequences of Ang II/AT1R stimulation, both AT2R, activated by Ang II, and MasR, triggered and internalized by Ang-(1-7) [24], exert cGMP- and NO-mediated vasodilation, diuresis and natriuresis (with the induction of natriuretic peptide [25]), and attenuate inflammation, oxidative stress, cell proliferation, apoptosis, and coagulation [3,7,22,26] (Figure 1). Furthermore, Patel et al., studying obese Zucker rats, found that MasR co-localizes with AT2R in proximal tubular cells, and that both are functionally interdependent in terms of stimulating NO and promoting diuretic/natriuretic responses [27]. Thus, AT2R and MasR exert comparable physiologic responses, mediated by similar downstream mechanisms, namely cGMP and nitric oxide. In fact, renal vasodilation, attributed to Ang-(1-7), has initially been credited to its binding to AT2 receptors, which have been regarded as selective AT2R agonists [28]. Moreover, the protective ACE/Ang II-AT2R and the ACE2/Ang-(1-7)/MasR axes likely co-stimulate each other. For instance, Ali et al. noticed that the deletion of AT2R decreases the expression of the beneficial ACE2/Ang-(1-7)/MasR and increases the deleterious ACE/Ang II/AT1R axis in mice fed on a high-fat diet [29]. In line with this notion, the chronic activation of AT2R increased renal ACE2 activity and attenuated AT1R function and blood pressure in obese Zucker rats [30].

It is noteworthy that, as detailed in depth elsewhere [31], ACE, ACE2 and Ang-(1-7) affect the generation, action, degradation and elimination of bradykinin through complex interwoven mechanisms that may also be involved in the physiological responses to RAS. As a result, at the bottom line, Ang-(1-7) seems to promote [32,33,34] and potentiate the vasodilatory effect of bradykinin [35], exerting vasorelaxation through the endothelium-dependent release of nitric oxide. As with RAS, bradykinin exerts opposing physiologic responses by binding to different receptors, and the bradykinin B2 receptors are those mediating vascular relaxation, in concert with MasR and AT2R [26,31] (Figure 1). Finally, there are still gaps of knowledge regarding the RAS system, and newly discovered components such as alatensins are still being studied [3]. Such gaps of knowledge may explain some paradoxical outcomes, as discussed below.

To conclude, the RAS comprises two distinct and counteracting arms: the harmful ACE/Ang II/AT1R “pressor” arm, and the tissue protective “depressor” arm, consisting of the ACE/Ang II/AT2R and ACE2/Ang-(1-7)/MarS axes (Figure 2). The integrity of the two arms enables salt and water preservation without the associated uncontrolled vasoconstriction, oxidative stress, and tissue injury and remodeling. As detailed below, recent findings related to COVID-19 disease and to renal disorders exemplify the outcome of a disrupted balance between the pressor and depressor arms of RAS.

## 3. COVID-19 Disease: An Archetype of Imbalanced RAS

The SARS-CoV-2 pandemic led to a wider recognition and a better understanding of the ACE2/Ang-(1-7)/MasR axis, since, on one hand, membrane-bound ACE2 serves as a unique viral receptor in target host cells during viral binding and dissemination, but on the other hand, it is also internalized and degraded by this process. As detailed elsewhere [4,5], the depletion of membranal ACE2 develops in the COVID-19 disease as the virus–ACE2 complex internalizes and is degraded following attachment. Furthermore, unopposed Ang II triggers ADAM-17 (shedase), which detaches ACE2 from the cell membranes, thus depriving the target organs of membrane-bound ACE2 along with enhanced circulating ACE2. Consequently, the production of Ang-(1-7) at the tissue level decreases, leaving the ACE/Ang II/AT1R axis unopposed. Conceivably, this leads to the intense inflammation, oxidative stress, tissue damage and coagulopathy that characterizes the COVID-19 disease. The incidence of AKI in this disorder may be as high as 46% in ICU patients with co-morbidities and requiring respiratory support [8]. As the complex pathogenesis of renal failure in patients infected with SARS-CoV-2 [36,37] likely includes the outcome of imbalanced renal RAS [38], it is tempting to assume that the strengthening of the depressor arm of the RAS by means of the administration of ACE2, Ang-(1-7) or other MasR agonists, or inhibiting the pressor arm, for instance with AT1R blockers, might restore the balance between the two arms of the RAS, attenuate disease severity, and perhaps provide renal protection [4,5].

## 4. ACE2/Ang-(1-7)/MasR Axis and Renal Physiology

The pressor and depressor arms of the RAS are extensively expressed in the kidney, and the altered ACE2/ACE balance characterizes renal dysfunction and participates in the progression of CKD [39]. In their thorough review of renal ACE2 [40], Lores et al. underscored the predominant expression of ACE2 in tubular epithelia, especially in the brush border of proximal tubules and the inner medullary collecting ducts, as well as in the vasa recta. ACE2 has also been detected to a much lower extent in vascular smooth muscle cells within the tunica media, and in the endothelium of renal arterioles, as well as in podocytes and mesangial glomerular cells [41,42,43]. As mentioned above, MasR and AT2R co-localize in the proximal tubular cells [27], but the distribution pattern of MasR within the renal microcirculation has not been evaluated. As to the location of renal vascular AT2R, Ang II infusion leads to total and cortical renal vasoconstriction in a dose-dependent manner, whereas medullary blood does not decrease [28] and may even increase, as shown in rodents [44,45,46]. This might reflect the diverse renal distribution of Ang II receptors, with the more prominent expression of AT2R in the vasa recta or juxtamedullary nephrons exerting medullary vasodilation [47], designed to maintain tissue integrity within the physiologically hypoxic medulla, in particular when the pressor arm of RAS is activated [48]. Of note, the detection and localization of intrarenal Ang-(1-7) is presently limited, since its determination by currently available Enzyme-Linked Immunosorbent Assay ELISA methods is inaccurate [49]. However, physiological concentrations of Ang-(1-7) were found to induce diuresis and natriuresis [50] and to exert the vasorelaxation of renal arterial ring segments pre-constricted with phenylephrine. This effect is endothelial-dependent, and involves NO and soluble guanylate cyclase [51]. It is noteworthy that Ang-(1-7)-mediated natriuresis is related not only to vasorelaxation and enhanced glomerular filtration, antagonizing the renal hemodynamic effects of Ang II [52], but also to the inhibition of sodium transport along the nephron [50,53,54], which is in part likely related to the enhanced generation of natriuretic peptide [25].

Studies in micro-dissected human glomerular and tubular samples disclose that renal ACE2 expression is gender-dependent, being more prominent among females [55]. Additionally, testosterone enhances AT1R expression in males, whereas estrogen preferentially upregulates AT2R and MasR expression [56,57] in females. It is tempting to assume that these sex-hormone-dependent differences might underlie the male susceptibility to hypoxic AKI, as noted under experimental settings [58,59,60]. Furthermore, aging is associated with declining ACE2 expression, which inversely correlates with the expression of mediators of inflammation, such as monocyte-chemoattractant protein-1 (CCL2, also known as MCP-1), interleukin (IL)-6 and tumor necrosis factor (TNF) [55]. ACE2 KO male mice (but not female mice), with or without chronic Ang II infusion, showed increased oxidative stress and molecular signals of inflammation and fibrosis with enhanced glomerulosclerosis, which has been prevented by ARB [61] or with recombinant ACE2 [62]. All these associations fit well with the concept of the organ-protective and anti-inflammatory roles of the ACE2/Ang-(1-7)/MasR axis.

## 5. Alterations in ACE2/Ang-(1-7)/MasR Axis in Systemic and Renal Disorders

Upon discovery of the ACE2/Ang-(1-7) axis, an altered ACE/ACE2 balance was studied in numerous clinical conditions affecting the kidney [63]. By and large, most were characterized by a reduced ACE2/Ang-(1-7) axis, promoting the predominance of the Ang II/AT1R axis. Not surprisingly, the first disease to be studied was hypertension, initially under experimental settings reported in a seminal study in *Nature* by Crackower et al. [64]. Indeed, as summarized by Lores et al. [40], the renal transcription and expression of ACE2 were suppressed in most animal models of hypertension, with a reciprocal enhancement of ACE, suggesting a role for the imbalanced pressor and depressor arms of RASS in this disorder. The progressive tubular depletion of ACE2 parallels the development of hypertension in the spontaneously hypertensive rat (SHR) strain, whereas glomerular expression increases, perhaps as a compensatory mechanism to enhance glomerular filtration [65]. ACE2 KO mice as well as MasR KO mice developed hypertension during the chronic infusion of Ang II, associated with renal fibrosis, which was most prominent in double ACE2/MasR KO animals [66]. The loss of ACE2 promotes and enhances hypertensive nephropathy induced by Ang II by targeting Smad7 for degradation [67]. ACE2 expression and activity increase in normal pregnancy in rats, both in the renal cortex and medulla, with increased renal Ang-(1-7), likely in response to increased Ang II levels. By contrast, Ang-(1-7) declined in hypertensive pregnant rats (induced by reduced uterine blood flow), as compared with normal pregnancy, even though the renal ACE2 activity remained unchanged, suggesting that other ANG-(1-7)-forming or degrading enzymes are involved [68].

Most importantly, the findings in humans parallel those in experimental settings: in cortical biopsies obtained from patients undergoing kidney biopsies, the ratio of renal cortical ACE/ACE2 expression was found to be higher in hypertensive individuals as compared with normotensives [69].

Renal ACE2 expression is also altered in various experimental and clinical conditions directly affecting the kidney. CKD leads to reduced renal ACE2 expression, as repeatedly shown in models of subtotal nephrectomy [70,71,72]. In humans, kidney biopsies obtained from patients with diabetic nephropathy also disclosed a reduced transcription of ACE2, both in glomeruli and in proximal tubules [73,74], whereas in an additional study in humans, CKD was associated with reduced glomerular, but not tubulointerstitial, ACE2 [55].

Altered renal ACE2 was also noted in experimental diabetes without overt nephropathy. Wysocki et al. reported increased ACE2 activity in cortical tissue without increased transcription in streptozotocin (STZ)-induced type 1 diabetic mice [75]. In another study, renal tubular ACE2 expression was markedly reduced in STZ-induced diabetes in mice [76]. However, in another study in STZ-induced diabetes and renal dysfunction in rats, ACE2 mRNA increased, but to a lesser extent than ACE, suggesting the predominance of the ACE/Ang II/AT1R axis [77]. In diabetic Akita mice, hyperglycemia was found to directly stimulate the ACE/Ang II/AT2R axis and to suppress ACE2 and MasR in proximal tubules through the enhanced factor erythroid 2-related factor 2 (Nrf2) activator [78]. By contrast, most studies in experimental type 2 diabetes, such as the report by Wysocki et al. in db/db mice [75], illustrate a substantial rise in tubular ACE2 expression and activity, though glomerular ACE2 declines. Possibly, differences in the severity of insulin deficiency, the magnitude of hyperglycemia and the duration of diabetes, affecting effective blood volume, renal morphology, hemodynamics, GFR status and sympathetic activity, may explain these differences. Lores et al. [40] also proposed a post-transcriptional regulation of ACE2 expression in diabetes from the perspective of inconsistency regarding ACE2 transcription across species, as well as non-concordant changes in ACE2 transcription and expression, in some of these studies. The impact of dietary salt may also be considered, as a high-salt diet was found to suppress the expression of the ACE2/AT2R/MasR axis in obese Zucker rats [79]. The human data regarding ACE2 are also inconsistent. Urinary expression of ACE2 mRNA in type 2 diabetic patients with nephropathy correlated with the degree of proteinuria, and inversely correlated with kidney function [80], in line with most experimental studies showing the upregulation of renal tubular ACE2. By contrast, the ACE2 protein and mRNA in the tubulointerstitium and glomeruli of type 2 diabetes subjects with overt proteinuria were reduced with an increased ACE/ACE2 ratio, proportional to renal dysfunction [73,74], suggesting ACE2 depletion with advanced diabetic nephropathy.

The importance of ACE2 in delaying the progression of CKD (via an AT1R-mediated mechanism) has been suggested in subtotal nephrectomized rats submitted to ACE2 inhibition [71], as well as in ACE2 KO Akita diabetic mice that exhibited a substantial acceleration of diabetic glomerulopathy and albuminuria [81]. In the same fashion, inflammation and fibrosis were substantially intensified in ACE2 KO mice subjected to unilateral ureteral obstruction [82].

As anticipated, heart failure also affects renal ACE2 expression. The expression of ACE2 was increased in a rat model of high-output heart failure, as long as the heart failure remained compensated, likely counterbalancing the enhanced ACE/Ang II/AT1R axis. By contrast, ACE2 levels declined in animals developing decompensated heart failure with salt and fluid retention, underscoring the importance of ACE2/Ang-(1-7)/MasR in maintaining natriuresis and diuresis in this disorder [83].

Renal hypoxia of whatever cause may also alter the regional balance between the two arms of the RAS through the stabilization and nuclear translocation of hypoxia inducible factors (HIF) [84]. HIF, a key regulator of the expression of multiple genes, plays an important role in tissue protection or maladaptation to acute or chronic renal hypoxia, respectively, the latter promoting inflammation and fibrosis [48,85]. HIF enhances the expression of ACE [86]. On the other hand, it suppresses ACE2, in part through the enhanced expression and activity of ADAM17 [87], which cleaves the cell membrane attachment of ACE2, leading to the depletion of ACE2 at the tissue level [5]. Collectively, an altered Ang II/Ang-(1-7) balance could lead to tissue hypoxia, with a feed-forward loop promoting HIF-mediated pro-inflammatory and pro-fibrotic changes.

Most of the studies detailed above address the tissue transcription, expression and activity of ACE2. Schmidt et al. recently studied the circulating plasma levels in patients undergoing kidney biopsies. They found higher levels among men, diabetics and patients with CKD, and reported the lack of an effect on circulating ACE2 with the use of ARBs and ACEi [88]. However, the origin of circulating ACE2 is evidently complex. Some is synthesized de novo, without its transmembrane domain [89], while the remaining is formed via the shedding of membrane-bound ACE2 by ADAM-17. The relative contribution of the two sources in health and disease, and in particular during COVID-19 disease progression, is not known. One can only point out the gender-related difference in the expression of circulatory and cell-bound ACE2 in humans, with higher renal ACE2 [55] but lower free circulatory ACE2 in females [88], and speculate that it indicates that shedding from tissues might be more predominant. The functionality of circulating ACE2 is also speculative. It may act on circulating Ang I and Ang II, but more likely, the membrane-bound isoform is the important one, acting on angiotensin-derived molecules at the tissue level.

## 6. Activating Renal ACE2/Ang-(1-7)/asR: Plausible Therapeutic Interventions in Renal Diseases

The administration of exogenous Ang-(1-7) or other agonists of MasR seems logical in the management of hypertension [90]. The same might be said regarding slowing the progression of CKD, and perhaps even in the management of AKI, preventing its transformation to CKD [91], from the perspective of the vasodilatory, antioxidant, anti-inflammatory, anti-apoptotic, and anti-fibrotic properties of this intervention. In fact, some of the renoprotective and blood-pressure-lowering effects of ARBs and ACEi have been attributed to the enhanced expression of ACE2 and the generation of Ang-(1-7) [70,92], and the same might be said to the impact of pregnancy on hypertension [93]. It is likely that the beneficial actions of ACE2 to a large extent reflect the accelerated degradation of Ang II, which is transformed into more Ang-(1-7) [94].

The exogenous administration of Ang-(1-7) is also an option and it is especially appealing in patients with severe COVID-19 disease, given the high incidence of AKI [95] in this ACE2-depleted disorder [4,5], and such clinical trials are currently under way [22,96]. As the bioavailability of ACE2 and of its downstream product Ang-(1-7) is low, novel compounds were developed in order to provide an extended action of Ang-(1-7), including ACE2 activators, MasR agonists, and stable Ang-(1-7) formulations [97]. For example, the administration of a chimeric ACE2–Fc fusion protein with prolonged bioavailability attenuated renal fibrosis in chronic hypertensive transgenic mice [98]. Regarding diabetic nephropathy, the chronic exogenous administration of Ang-(1-7) in Akita diabetic mice attenuated hypertension, oxidative stress, and progressive features of renal functional and morphological damage, and in a feed-forward loop enhanced ACE2 and MasR expression [99]. Similarly, cyclic (c)Ang-(1-7), a lanthipeptide that is more peptidase-resistant than the linear peptide, was found to attenuate the progression of diabetic nephropathy in ob/ob mice with type 2 diabetic nephropathy [100]. Recombinant human ACE2 exerted comparable protection in Akita diabetic rats with reduced Ang II and enhanced Ang-(1-7) signaling [101]. Meems et al. engineered and studied a novel bispecific designer peptide (NPA7), consisting of fused BNP and Ang-(1-7) (activating both GC-A and MarsR, respectively), with substantial natriuresis and diuresis and with renal and systemic vasodilatory properties [102]. In a mouse model of progressive glomerulopathy induced by Adriamycin, Barroso et al. reported attenuated proteinuria and glomerular injury with a MasR agonist AVE 0991 [103]. Interestingly, this model is associated with the reduced transcription of MasR, which was restored with ARBs. In order to stimulate the ACE2/Ang-(1-7)/MasR axis specifically at its most intense site of renal expression, at the proximal tubular brush border, Lores et al. proposed the administration of ACE2 or smaller active ACE2 derivatives (enabling filtration through the glomerular slit diaphragm) under conditions characterized by reduced ACE2 expression and activity, as in diabetic and hypertensive renal disease, and possibly in COVID-19 disease [40].

Other interventions aimed at the amelioration of renal injury may indirectly stimulate the ACE/Ang II/AT1R axis. Renal sympathetic denervation is one such strategy, as it was found to suppress and upregulate the expression of both ACE2 and MasR in hypertensive rats [104]. Antioxidant treatment may work in the same fashion. In aged rats, chronic administration of the antioxidant resveratrol enhanced the renal expression of ACE2, AT2R and MasR, in parallel with reducing oxidative stress, whereas the renal ACE/Ang II/AT1R axis became suppressed [105]. Tempol was also found to enhance the expression of renal tubular ACE2, AT2R and MasR, as well as intensifying renal parenchymal Ang-(1-7) in rats fed on a normal or high-salt diet [106]. Inhibiting Nrf2 in diabetic mice also resulted in the enhanced expression of ACE2 and MasR and increased urinary Ang-(1-7) [78]. Finally, suppressing the ACE/Ang II/AT1R axis with ARBs or ACEi causes a reciprocal enhancement of ACE2 activity in the renal cortex and of plasma Ang-(1-7) [107]. Thus, an altered balance between the pressor and protective arms of RAS can also be restored in chronic and perhaps acute renal injury indirectly, by restoring or stimulating MasR.

## 7. Enhancing the ACE2/Ang-(1-7)/MasR Axis in the Management of AKI

The beneficial impact of exogenous MasR stimulation in the management of experimental CKD let to the adoption of this approach in the setup of AKI. The impact of ACE/ACE2 imbalance in the setup of acute renal injury has been evaluated in a mouse model of rhabdomyolyisis–AKI, induced by protracted hindlimb ischemia [108]. This model leads to an increased renal ACE expression, while renal ACE2 generation decreases, associated with elevated serum Ang II level and lowered serum Ang-(1-7). However, both renal Ang II and Ang-(1-7) were found to be increased in this setup, the latter conceivably reflecting a physiological attempt to attenuate renal injury. This model was further explored in ACE2 knock-out (KO) and in ACE2-trangenic mice. In ACE2 KO mice, renal injury was significantly aggravated, associated with an upregulation of ACE, as compared with wild-type mice. Conversely, ACE2 transgenic mice with normal ACE expression were more resistant to rhabdomyolysis AKI, with attenuated renal pathological changes and increased survival rate [108].

Rats subjected to renal warm ischemia and reperfusion (WIR-AKI) also display an altered AngII/Ang-(1-7) balance. In rats subjected to uninephrectomy and 45’ ischemia, the renal Ang II increased and Ang-(1-7) declined at 2–4 h following reperfusion. However, the mRNA expression of MasR markedly increased [109]. In another study, WIR was shown to increase plasma Ang II, while plasma Ang-(1-7) declined. A parallel changing pattern of Ang species was found in renal proximal tubules [110]. These changes, more prominent in streptozotocin-induced diabetic rats, reflect an upstream alteration in the synthetic machinery, i.e., the enhanced expression of ACE in proximal tubules and a reciprocal decline in ACE2 following renal WIR [110]. Furthermore, renal WIR was also associated with an upregulation of renal AT1R, AT2R and MasR. The administration of an oral ACE2 activator, diminazene aceturate, or an AT2R agonist (compound 21), and especially their combination, was associated with a rise in renal tubular ACE2 expression and Ang-(1-7), and with an increased detection of AT2R and MasR. Most importantly, these agents, and especially their combination, significantly attenuated renal morphologic injury in diabetic rats subjected to WIR, with an attenuation of kidney dysfunction [110]. A reduced inflammatory reaction following WIR with improved renal function and morphology was also achieved by the administration of AVE0991, an oral MasR agonist to MasR KO mice [111].

In another WIR-AKI study in non-diabetic ACE2 KO mice [112], the extent of neutrophil, macrophage, and T cell infiltration within the kidney increased, as compared with wild-type (WT) mice, along with the enhanced mRNA expression of pro-inflammatory cytokines (IL-1β, IL-6 and TNF) and chemokines (macrophage inflammatory protein-2 and monocyte chemoattractant protein-1). There was a greater extent of apoptosis and oxidative stress in the ACE2 KO mice, as compared to WT animals, yet the histologic injury scores and measures of kidney function at 48 h after reperfusion were comparable [112]. The administration of the ACE2 activator diminazene aceturate was associated with a rise in kidney nitrites, as well as attenuated renal dysfunction and indices of oxidative stress, in male rats subjected to WIR. Its lack of efficacy in female rats subjected to WIR [113] might reflect gender differences in the expressions of ACE2, MasR and AT1R between species [114].

The ACE2 activator diminazene aceturate also blunted the rise in TNF in a rat model of nephrotoxic AKI, induced by gentamycin, and attenuated renal functional derangement (rising creatinine, proteinuria and polyuria) and morphologic injury, suppressing the extent of the inflammatory infiltrate [115]. In a rat model of unilateral ureteral obstruction, the continuous administration of Ang-(1-7) also attenuated the extent of tubular cell apoptosis, and mitigated fibrosis [116].

An intensification of the ACE2/Ang-(1-7)/MasR axis in AKI could take place indirectly, for instance with antioxidants, as shown in CKD. Another documented example is the restoration of ACE2 transcription, reduced in experimental sepsis AKI, managed by activated protein C [117].

There are as yet no data regarding the impact of restoring the ACE/ACE2 balance in COVID-19 with or without AKI. An initial concern regarding the use of ACEi and ARBs has been based on the induction of ACE2 with these medications, facilitating viral attachment to host target cells [4]. However, further debate has brought about the operative conclusion that as ARBs and ACEi attenuate the Ang II/AT1R pressor arm, which may compensate the depletion of ACE2 and Ang-(1-7) and help maintain the balance between the opposing arms of RAS, their use should not be interrupted [118]. As stated before, current studies are being conducted to assess the systemic and renal protective potential of stimulating the ACE2/Ang-(1-7)/MasR axis in COVID-19 patients.

Taken together, the in vivo studies detailed above demonstrate the protective effect of the ACE2/Ang-(1-7)/MasR axis in various forms of nephrotoxic and hypoxic AKI, and form a conceptual basis for the activation of this axis as a therapeutic intervention in human AKI. The renoprotective mechanisms considered to be activated during the stimulation of the ACE2/Ang-(1-7)/MasR axis might include the alleviation of hypoxic stress in hypoxic AKI by the restoration of renal microcirculation and the decrease in oxygen consumption for tubular transport. This may also attenuate the inflammation generated in sepsis AKI and in various forms of nephrotoxic AKI, and may also alleviate oxidative stress, apoptosis and post-AKI pro-fibrotic processes, shared by all types of AKI. Yet, as outlined below, the evidence for renal salvage with this strategy is not fully consistent, suggesting that other detrimental physiologic responses may be activated as well under certain conditions.

## 8. Enigmatic Contradicting Findings

Despite the overall promising reports regarding the beneficial potential of the stimulation of the ACE2/Ang-(1-7)/MasR axis, there are a few reports calling for caution, indicating the potential paradoxical adverse renal outcomes under experimental settings of chronic and acute renal injury. For example, Estaban et al. [119] found less renal damage following WIR or unilateral ureteral obstruction in MasR KO mice, as compared with WT animals. Furthermore, Ang-(1-7) infusion had proinflammatory effects in these experimental models of renal failure, as well as under basal conditions. Burrell et al. [120], summarizing their unexpected contradicting findings in rats with subtotal nephrectomy and a few previous studies, suggest that this paradoxical outcome might reflect the enhanced production of Ang II. They report that the concomitant administration of ACEi in conjunction with Ang-(1-7) abolishes injury invoked by Ang II and consolidate the beneficial impact of Ang-(1-7). Bi et al. [121] reached comparable conclusions, studying awake uninephrectomized rams with antenatal betamethasone exposure. They found that Ang-(1-7), administered directly into the renal artery, provoked renal vasoconstriction and reduced natriuresis, a response characterizing AT1R stimulation, which was reversed by ARBs. They proposed that the prenatal administration of glucocorticoids alters the expression pattern of the various mediators and receptors of the angiotensin family. Their additional experiments using MasR and AT1R blockers imply that this paradoxical response may reflect Ang-(1-7) activating AT1R, particularly in hypertrophied overactive remnant nephrons [121]. Possibly, such unexpected findings, which contradict most studies, may reflect the actions of as-yet unknown or partially understood components, such as alatensins [3], in the general scheme of RAS. This idea is also supported through work undertaken by Zimmerman et al., who found that Ang-(1-7) further increased the apoptosis and macrophage influx in obstructed kidneys from MasR-deficient mice, indicating that part of the detrimental effect of the heptapeptide is MasR-independent and might relate to a receptor that is especially upregulated in this experimental scenario of unilateral ureteral obstruction [122]. Furthermore, the various descriptions of the detrimental effects of exogenous Ang-(1-7) might also be caused by an altered interplay between RAS components, such as an upregulation of Ang II/AT1R, and/or a depleted ACE2/MasR induced by continuously increased Ang-(1-7) [123]. An altered Ang II/Ang-(1-7) ratio may also cause gender-dependent diverse expressions of renal T cell sub-populations, affecting the cellular response to injury, as shown in rodents chronically infused with Ang II [124]. Additional explanations for such discrepancies in the WIR model might be technical (i.e., differences in species and strains and in the perturbations and AKI techniques applied), or advanced disruption of the ACE2/Ang-(1-7)/MasR pathway, which should be especially damaged in WIR, with extensive necrosis of the S3 segments of the proximal tubules [1,2]. Other paradoxical outcomes, such as reduced renal fibrosis in ACE2 KO diabetic mice [76], possibly reflect the diverse impact of glomerular and tubular ACE2 on renal oxygenation. On the one hand, blocking glomerular ACE2 may attenuate hyperfiltration, with consequently reduced tubular oxygen consumption for sodium transport and improved renal oxygenation. Additionally, the inhibition of tubular ACE2 might enhance proximal tubular sodium reabsorption, leading to reduced cortical oxygenation, as opposed to the inhibition of tubular transport, for instance with SGLT2 inhibitors, which improves cortical oxygenation [125]. Of note, the impact of acute or chronic stimulation of the ACE2/Ang-(1-7)/MasR axis on renal oxygenation and microcirculation in intact and injured kidneys has not been studied so far. Most importantly, Burns et al. [126] provide evidence for the induction of epithelial-to-mesenchymal transformation (EMT) by the ACE2/Ang-(1-7)/MasR pathway. This could show that cultured tubular cells subjected to Ang II underwent EMT, which was blocked by a MasR antagonist, but not by an AT1R antagonist. Moreover, Ang-(1-7), an MasR agonist, and recombinant ACE2 were all able to induce EMT, and in vivo renal fibrogenesis was induced by chronic Ang-(1-7) infusion [126].

Thus, there is substantial evidence that the stimulation of the ACE2/Ang-(1-7) MasR axis, especially when applied chronically, may be harmful under certain conditions via a complex of partially revealed mechanisms. Perhaps this strategy might be safer in the management of CKD or AKI with the concomitant inhibition of the ACE/Ang II/AT1R axis [127].

## 9. Conclusions

The ACE/Ang II/AT1R pressor axis exerts renal vasoconstriction, inhibits natriuresis and diuresis, and promotes oxidative stress, tissue inflammation, apoptosis and subsequent fibrosis. Blocking this axis by ACEi and ARBs has become a standard of care in the management of CKD, with unequivocal clinical evidence for the preservation of renal integrity and function along time. The ACE2/Ang-(1-7)/MasR depressor arm serves as an inbuilt counterbalancing system that controls the action of the pressor arm, and its exogenous stimulation might provide comparable renal protection, when administered alone, and all the more so when combined with ACEi or ARBs. From that perspective, recent findings indicate that activating this system might be of value in the management of hypertension, and may be renoprotective in the setup of CKD, as well as in AKI. However, there are large knowledge gaps regarding the impact of the ACE2/Ang-(1-7)/MasR axis on renal microcirculation, tubular metabolic demands and renal oxygenation gradients. With that in mind, and with contradicting reports regarding the renal response to Ang-(1-7) under normal settings and in the diseased kidney, additional confirmatory studies are needed before the adoption of this strategy in humans with AKI. The COVID-19 disease might be an exception to this conclusion, with or without AKI, given that it is characterized by the depletion of tissue ACE2, and that its clinical features suggest an imbalanced ACE/ACE2 ratio.

## Figures and Tables

**Figure 1 jcm-10-01200-f001:**
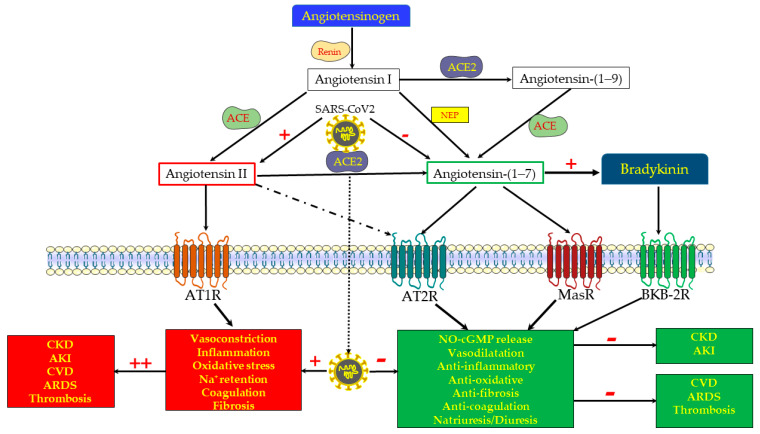
Biosynthesis and functional scheme of the renin–angiotensin system (RAS). The classical RAS consists of the protease renin, which is secreted from renal juxtaglomerular cells adjacent to the afferent arteriole, and which acts on the circulating precursor angiotensinogen to generate angiotensin (Ang) I, an inactive 10 amino acid (aa) peptide. The latter is converted by the angiotensin-converting enzyme (ACE) to Ang II, an 8 aa active peptide. Ang II is the main effector component of the RAS, as evident from its potent action, stimulating vasoconstriction, oxidative stress, Na+ retention, inflammation, fibrosis and coagulation, all mediated by the AT1 receptor (AT1R). However, Ang II also acts via AT2R, which is part of the “depressor” or protective arm of RAS, as made evident by its oppositely effects on the various target organs, including the kidney, heart, and vasculature. ACE2 is a peptidase located on cell membranes in various tissues, including the kidneys, where it promotes the proteolytic cleavage of the octapeptide Ang II with the formation of Ang-(1-7). Alternatively, Ang 1-7 can be generated by a preliminary cleavage of the decapeptide Ang I with the formation of Ang 1-9, followed by the removal of two additional amino acids by ACE. Ang-(1-7) may also be formed directly from angiotensinogen, following proteolytic cleavage by neprilysin (NEP). Ang-(1-7) acts via the Mas receptor (Mas-R) to stimulate nitric oxide (NO) and cGMP, exerting vasodilation and attenuating inflammation, oxidative stress, pro-fibrotic processes, coagulopathy, and probably permeating diuresis and natriuresis. Ang-(1-7) also modifies the kinin pathways, promoting bradykinin action through its type B_2_ receptors. Since AT2R and MasR exert comparable physiologic responses, mediated by the same downstream mechanisms, namely cGMP and NO, it is assumed that the Ang-(1-7) actions are also attributed to its binding to AT2R (a receptor activated also by Ang II). Thus, the ACE2/Ang-(1-7)/MasR + AT2R axis forms an under-recognized beneficial arm of the RAS, that in concert with bradykinin/bradykinin type B_2_ receptors counterbalances the delirious arm, namely the ACE/Ang II/AT1R axis, which is involved in the pathogenesis of various cardiovascular, pulmonary, renal and hematological diseases. The COVID-19 disease is characterized by the depletion of ACE2 and Ang-(1-7) along unleashed ACE/Ang II/AT1R, conceivably playing a central role in the devastating cytokine storm, oxidative stress, coagulopathy and fibrosis that characterizes this disorder. In light of their beneficial effects on the progression of cardiovascular, pulmonary and renal diseases, ACE2 repletion and the administration of Ang-(1-7) constitute the therapeutic options currently tested in the management of severe COVID-19 disease, with the expectation of renal vasodilation and the attenuation of oxidative stress, inflammation, tissue damage and subsequent fibrosis. Although most outcomes indicate that triggering the ACE2/Ang-(1-7)/MasR axis may be nephroprotective in the setup of AKI, there is contradicting evidence that under certain conditions, it may accelerate renal damage in CKD and AKI.

**Figure 2 jcm-10-01200-f002:**
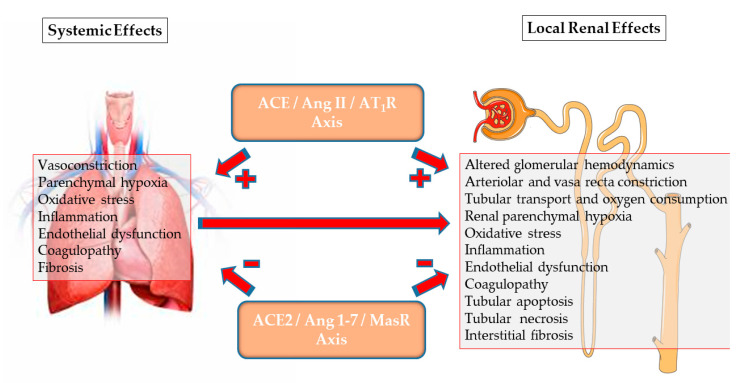
Systemic and direct renal effects of the two arms of RAS leading to acute kidney injury (AKI): the pressor and harmful arm, mediated by the ACE/Ang II/AT1R axis, and the counteracting depressor and protective arm, mediated by the ACE2/Ang-(1-7)/MasR axis and Ang II/AT2R. In addition to a direct impact on the kidney, systemic RAS imbalance, as occurs in COVID-19 disease, can participate in acute renal dysfunction and injury, which may progress to CKD.

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
