# Peer review of "Angiotensin-(1-7)—A Potential Remedy for AKI: Insights Derived from the COVID-19 Pandemic"

_jcm, 2021, doi:10.3390/jcm10061200_

Round 1

Reviewer 1 Report

The submitted manuscript is a review article discussing the role of ACE2 and Angiotensin-(1-7) - a Potential Remedy for AKI: Insights De-2 rived from the COVID-19 Pandemia.  

Major Concerns

  1. Overall manuscript seems to repeat the benefits of ACE2/Ang1-7 including vasodilation, diuresis and natriuresis and attenuation of inflammation, oxidative stress, cell proliferation, apoptosis, and coagulation, at multiple points in the beginning, without giving the concise mechanisms. For the purpose of this article, although the references are given, a short summary of the downstream pathways of these beneficial effects would be useful.
    1. For example, line no 127-130 talk about bradykinin induced vasodilation. Elaborate if bradykinin is the only or one of the mechanisms by which Ang 1-7 causes vasodilatation.
  2. The literature is not consistent regarding AT2R Involved in Ang-(1–7) action. However, the figure 1, there is arrow Ang 1-7 to AT2R- clarify that in the text. Ref. (Hypertension. 2014;63:1138-1147.)
  3. Authors describe that Ang II triggers ADAM-17 and detachment of ACE2 from the cell membrane resulting in high circulating ACE2. What is the utility or functionality of circulating ACE2 as compared to membrane bound – please describe that.
  4. The paragraph from line 187-200 was discussing the gender difference of ACE2 but conclusion ends talking about the organ-protective role of ACE2/Ang1-7 axis, without concluding gender difference.
  5. Line 204-205. Should it be ACE2/Ang1-7 ratio or absolute values of this axis. As described by authors that in most conditions, there is reduced ACE2/Ang1-7 ratio – does that mean ACE2 levels low but high Ang 1-7 levels to cause ratio to be low. Later in the same paragraph – authors mention that in hypertensive pregnant rats, Ang 1-7 decline with unchanged ACE2 – that will cause ratio to be high.
  6. Line 204. It states exception of experimental diabetes; however, later altered activities of ACE2/Ang 1-7 are described in the experimental model of diabetes.
  7. Line 258-262. ACE2 inhibition or knock out causing worsening disease doesn't mean that ACE2 administration would delay the progression. Did these studies replace or increase the ACE2 and show the delayed progression? Only in the latter case, the conclusion in the first sentence of the paragraph can be supported.
  8. Line 244-245. Provide the reference for the assumption.
  9. Advise to provide summary statement for the paragraphs under heading # 5 and 6, since too many studies have different/contradictory observations.

Minor Concerns:

  1. Line 151. It should read COVID-19, not -2, COV2 is 2 but not the disease.
  2. Line 194. monocyte-chemoattractant protein-1 (CCL2), the short form is wrong, should be MCP-1.
  3. Line 205-206. The sentence is incomplete and incomprehensible.
  4. Line 285. Revise the language. “thanks to the vasodilatory, antioxidant……….”, it is not a standard manuscript language.

Reviewer 2 Report

I'd like to congratulate the authors on putting together a comprehensive review on ACE2/Ang1-7 signaling in the context of kidney disease.  There is an array of research covered evaluating different component of the Ang1-7/MasR signaling cascade and ACE2 in the context of a number of different kidney diseases.  The implication of these previous research on the treatment of AKI associated with COVID-19 is alluded to in this review. Two minor components that can be included.

  • The discussion of ACE2/Ang1-7/MasR axis in renal physiology was helpful but it maybe helpful to highlight the natriuresis seen is not strictly dependent on increased glomerular filtration but with inhibition of tubular sodium transport as suggested by reference 48 and also the additional references cited here. (DOI: 10.1152/ajprenal.1996.270.1.F141, DOI: 10.1038/ki.1993.334)

  • The recent evaluation of plasma ACE2 in the context of CKD by Schmidt et al. provides some human data in this capacity and maybe a helpful citation to include under subheading 5. Alterations in ACE2/Ang-1-7/MasR axis in systemic and renal disorders: (https://doi.org/10.1093/eurheartj/ehaa523)

Round 2

Reviewer 1 Report

Thanks for replying to the concerns.